# Pre-40S ribosome biogenesis factor Tsr1 is an inactive structural mimic of translational GTPases

Urszula M. McCaughan[1], Uma Jayachandran[1], Vadim Shchepachev[1], Zhuo Angel Chen[1], Juri Rappsilber[1,2], David Tollervey[1] & Atlanta G. Cook[1]

Budding yeast Tsr1 is a ribosome biogenesis factor with sequence similarity to GTPases, which is essential for cytoplasmic steps in 40S subunit maturation. Here we present the crystal structure of Tsr1 at 3.6 Å. Tsr1 has a similar domain architecture to translational GTPases such as EF-Tu and the selenocysteine incorporation factor SelB. However, active site residues required for GTP binding and hydrolysis are absent, explaining the lack of enzymatic activity in previous analyses. Modelling of Tsr1 into cryo-electron microscopy maps of pre-40S particles shows that a highly acidic surface of Tsr1 is presented on the outside of pre-40S particles, potentially preventing premature binding to 60S subunits. Late pre-40S maturation also requires the GTPase eIF5B and the ATPase Rio1. The location of Tsr1 is predicted to block binding by both factors, strongly indicating that removal of Tsr1 is an essential step during cytoplasmic maturation of 40S ribosomal subunits.

[1] Wellcome Trust Centre for Cell Biology, University of Edinburgh, Michael Swann Building, Max Born Crescent, Edinburgh EH9 3BF, UK. [2] Department of Bioanalytics, Institute of Biotechnology, Technische Universität Berlin, 13355 Berlin, Germany. Correspondence and requests for materials should be addressed to A.G.C. (email: atlanta.cook@ed.ac.uk).

Ribosome biogenesis in eukaryotes is a highly complex process that requires more than 200 non-ribosomal factors. This striking complexity may help ensure that ribosome production is both fast and accurate[1–4]. At a relatively early stage of processing, pre-40S particles are separated from pre-60S particles by a series of nuclease cleavage events and rapidly exit the nucleus[5]. The pre-40S particles that enter the cytoplasm are close to the mature structure but carry a specific complement of seven biogenesis factors[6,7]. These include structural proteins Enp1 and Ltv1, the methyltransferase Dim1, the protein kinase Rio2, the endonuclease Nob1 together with its binding partner Pno1 and the GTPase-related protein Tsr1. While ribosome biogenesis is best studied in *Saccharomyces cerevisiae*, these nucleo-cytoplasmic pre-40S biogenesis factors are also conserved in mammals[8].

The release of biogenesis factors generates structural and chemical changes in pre-40S particles, which are essential for its maturation and function in translation. However, the exact order of events during cytoplasmic maturation and the roles of individual factors are not fully understood. Analyses of 20S pre-rRNA structure and pre-40S particle composition have revealed a time course for the release of biogenesis factors during cytoplasmic maturation. 'Early' released factors include Enp1 and Ltv1, 'intermediate' factors include Rio2, Dim1 and Tsr1 while 'late' factors are Nob1 and Pno1 (refs 9–11).

A key final event in pre-40S maturation is the cleavage of 20S pre-rRNA to mature 18S rRNA by Nob1. Cleavage of 20S is stimulated by both ATP and GTP *in vitro*, indicating that hydrolytic enzymes drive late maturation events[12]. The translation initiation factor eIF5B (Fun12p in yeast) participates in pre-40S maturation by stimulating formation of an 80S-like particle comprised of pre-40S and mature 60S ribosome subunits[12,13]. Disruption of eIF5B binding to 60S subunits inhibits processing of 20S to 18S rRNA, indicating that formation of the 80S-like particle helps drive this late processing event[12]. Stimulation of pre-40S maturation in response to ATP is mediated by the non-canonical protein kinase Rio1, which associates with late pre-40S particles in the cytoplasm and may also promote formation of 80S-like particles[10].

Twenty-S rRNA accumulation 1 (Tsr1) is an essential 40S subunit biogenesis factor with homology to characterized GTPases[6–8,14]. Yeast depleted for Tsr1 accumulate 20S rRNA, showing that Tsr1 functions before pre-rRNA cleavage by Nob1 (ref. 14). Pre-40S particles purified by tandem affinity purification, with tagged Tsr1 show an rRNA structure profile (defined by chemical probing) consistent with early and intermediate cytoplasmic pre-40S particles, rather than later pre-40S ribosomal subunits[9]. Tsr1 shows sequence similarity to the small GTPase family, however, neither GTP binding nor GTPase activity has been detected *in vitro*[12,14].

To gain a deeper insight into the function of Tsr1 in ribosome biogenesis and to discover whether this protein is a functional GTPase, we determined the crystal structure of yeast Tsr1. Our structural data, combined with fitting into cryo-electron microscopy maps of pre-40S particles indicate that Tsr1 guards key sites in pre-40S particles from factors required for late maturation steps.

## Results

**Tsr1 is made up of four domains and a dynamic loop.** *S. cerevisiae* Tsr1 was expressed as an N-terminal hexahistidine-tag fusion protein in *Escherichia coli*. Attempts to crystallize the purified full-length protein failed. However, using limited proteolysis and size-exclusion chromatography, we identified two

stable fragments of Tsr1 that co-elute at a position similar to the undigested protein. Comparing Tsr1 sequences from a variety of eukaryotic organisms revealed the presence of a poorly conserved region that contains mainly acidic residues (Supplementary Fig. 1). We reasoned that this segment might hinder crystallization, so we generated a 'loop-out' mutant (Tsr1$_{\Delta loop}$), in which residues 410–476 of Tsr1 were replaced with a short glycine- and serine-rich sequence (Fig. 1a). Crystals of Tsr1$_{\Delta loop}$ diffracted to low resolution. A further deletion of sequence predicted to be natively unstructured at the N terminus (Tsr1$_{\Delta N \Delta loop}$) improved diffraction quality, generating native crystals that diffracted to 3.6 Å (Table 1).

The loop segment of Tsr1 that was deleted is not well conserved in size or in sequence across eukaryotes. However, the loop position is conserved, as is the high frequency of aspartate and glutamate residues in this segment. We found that Tsr1$_{\Delta N \Delta loop}$ complements loss of wild-type Tsr1 in yeast (Fig. 1b), showing that the N terminus and acidic loop are dispensable for Tsr1 function *in vivo*.

The crystal structure of Tsr1$_{\Delta N \Delta loop}$ was determined using a combination of low-resolution *de novo* phasing of a crystal derivatized with a tantalum bromide cluster (TaBr) combined with anomalous differences for intrinsic sulfur atoms. Model building and refinement with data from a native crystal produced a model with good stereochemistry (Table 1). Of the four molecules in the asymmetric unit, molecules A and C are the best defined at the N and C termini, respectively (Supplementary Fig. 2a,b). As it is the most complete model, we will focus our discussion on molecule A unless otherwise indicated. Comparison of the four molecules in the asymmetric unit shows only limited differences between them (root mean squared deviation (r.m.s.d.) values range from 0.9 to 1.1 Å) suggesting that the overall domain arrangement of Tsr1 is fixed (Supplementary Fig. 2b). Similarly, Tsr1 shows an excellent fit to a previously described structural envelope generated by negative-stain electron microscopy (EM; Supplementary Fig. 2c)[15].

Tsr1 is characterized by four domains that pack together to form a chalice-like structure (Fig. 1c). The N-terminal domain (domain I) has a small GTPase-like fold. The second, third and fourth domains (II–IV) are β-barrels. Domain II has a large insertion segment that wraps around the centre of the molecule. This insertion lacks secondary structure elements and is also the location of the deleted acidic loop. Parts of this insertion participate in intermolecular crystal contacts and are relatively well defined (residues 344–363 and 480–500, Supplementary Fig. 2d). Segments encompassing residues 312–335 and 380–479 within the insertion are not visible in the map. Further regions that are poorly defined in all molecules include residues 50–67, 117–120, 364–369 and 761–764. The first and last β-strands of domain IV form an extended, two-stranded β-sheet, placing domain IV at the periphery of the structure and burying the C terminus of the protein in the core of the protein between domains I and III (Fig. 1c).

To determine whether the internal deletion in Tsr1 affected its structure in solution, we generated intra-molecular crosslinking maps for full-length Tsr1 and Tsr1$_{\Delta N \Delta loop}$ using a crosslinking and mass spectrometry approach (Supplementary Fig. 3a, Supplementary Methods and Supplementary Data Set 1 and 2). Full-length Tsr1 and the double deletion mutant showed good agreement in their crosslinking patterns, further supporting the conclusion that the global structure is unaffected by the internal deletion. Furthermore, the majority of crosslinks in both samples include residues that were not visible in the electron density map. This gives complementary structural information on highly dynamic regions. For example, the majority of crosslinks found only in the full-length protein are between lysine residues in the

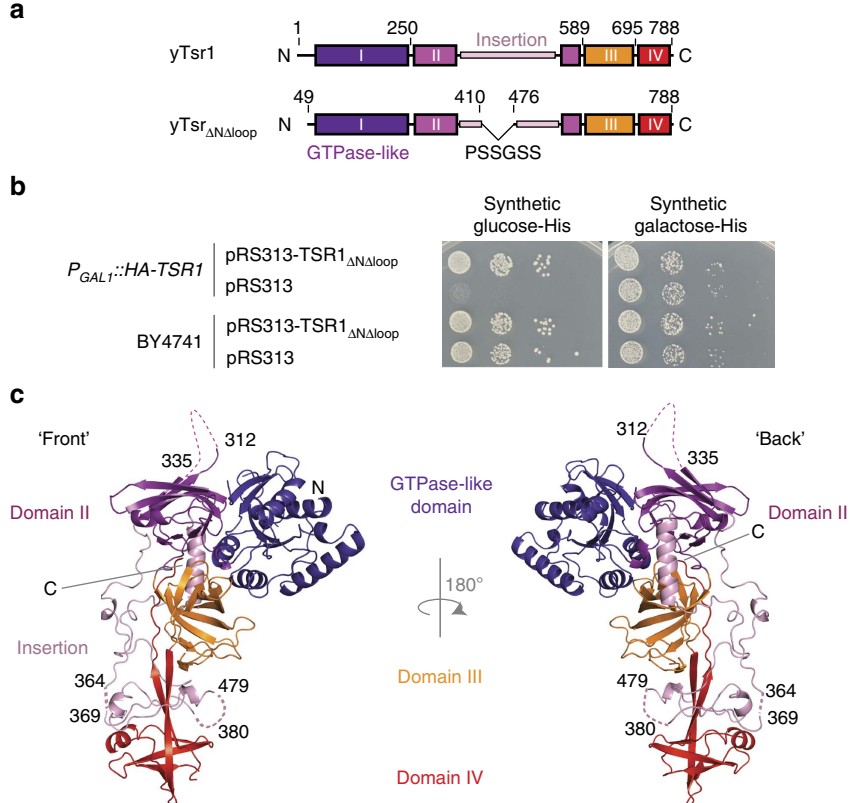

**Figure 1 | Tsr1 is a multi-domain structure.** (**a**) Schematic overview of Tsr1 domains, comparing the full-length protein with the Tsr1$_{\Delta N \Delta loop}$ deletion mutant used in crystallization. (**b**) *TSR1*(BY4741) and *pGAL::HA-TSR1* strains were transformed with plasmid pRS313 expressing the Tsr1$_{\Delta N \Delta loop}$ deletion mutant or the empty vector. Following growth in galactose medium, serial dilutions of the cultures were plated on galactose medium or on medium containing glucose to repress chromosomal expression of HA-Tsr1. (**c**) Overview of the structure defining front and back views, related by a 180° rotation. Domains are coloured according to the schema in **a**. Disordered regions are indicated with dotted lines and residue numbering.

**Table 1 | Data collection and refinement statistics.**

| Data collection | Native | TaBr cluster derivative MAD | | | Sulfur SAD (20 data sets from 11 crystals) |
| --- | --- | --- | --- | --- | --- |
| | | **Peak** | **Inflection** | **Remote** | |
| Space group | $P2_12_12_1$ | | $P2_12_12_1$ | | $P2_12_12_1$ |
| Cell dimensions, *a, b, c* (Å) | 65.7, 173.9, 319. 8 | 65.0, 175.2, 319.2 | 65.8, 175.4, 318.0 | 65.5, 175.4, 318.0 | 66.2, 174.4, 321.8 |
| *α, β, γ* (°) | 90, 90, 90 | 90, 90, 90 | 90, 90, 90 | 90, 90, 90 | 90, 90, 90 |
| Resolution (Å) (high-res. shell) | 48.8 − 3.6 | 48.6 − 4.2 | 48.6 − 4.2 | 48.6 − 4.2 | 50.0 − 3.9 |
| | (3.8 − 3.6) | (4.4 − 4.2) | (4.4 − 4.2) | (4.4 − 4.2) | (4.1 − 3.9) |
| Reflections | 437,274 | 143,710 | 143,704 | 134,224 | 6,878,352 |
| Unique reflections | 43,821 | 27,613 | 27,756 | 26,056 | 65,471 |
| $R_{meas}$ (%) | 12.7 (92.2) | 18.5 (54.3) | 18.9 (67.1) | 18.1 (62.1) | 35.3 (147.7) |
| CC(1/2) | 99.9 (89.5) | 99.6 (82.0) | 99.3 (72.8) | 99.4 (79.1) | 99.9 (89.7) |
| Completeness (%) | 99.9 (99.5) | 99.0 (94.4) | 99.3 (96.7) | 99.2 (92.8) | 99.9 (99.8) |
| $I/\sigma I$ | 14.4 (2.9) | 10.3 (3.2) | 9.5 (2.7) | 9.4 (2.6) | 24.8 (4.3) |
| Multiplicity | 10.0 (9.9) | 5.2 (5.3) | 5.2 (5.2) | 5.2 (4.9) | 105.1 (55.1) |
| Anomalous completeness | — | 97.0 (91.0) | 97.0 (92.9) | 96.9 (86.8) | 99.9 (99.8) |
| | | | | | |
| *Refinement* | | | | | |
| $R_{work}/R_{free}$ | 26.8/29.5 | | | | |
| r.m.s. deviations | | | | | |
| Bonds lengths (Å) | 0.007 | | | | |
| Bond angles (°) | 0.99 | | | | |
| Total number of protein atoms | 16,065 | | | | |
| Average B factor (Å$^2$) | All, 87.0 | Chain A, 50.1 | Chain B, 98.7 | Chain C, 91.7 | Chain D, 121.2 |

High res., high resolution; MAD, multiple wavelength anomalous dispersion; SAD, single wavelength anomalous dispersion.

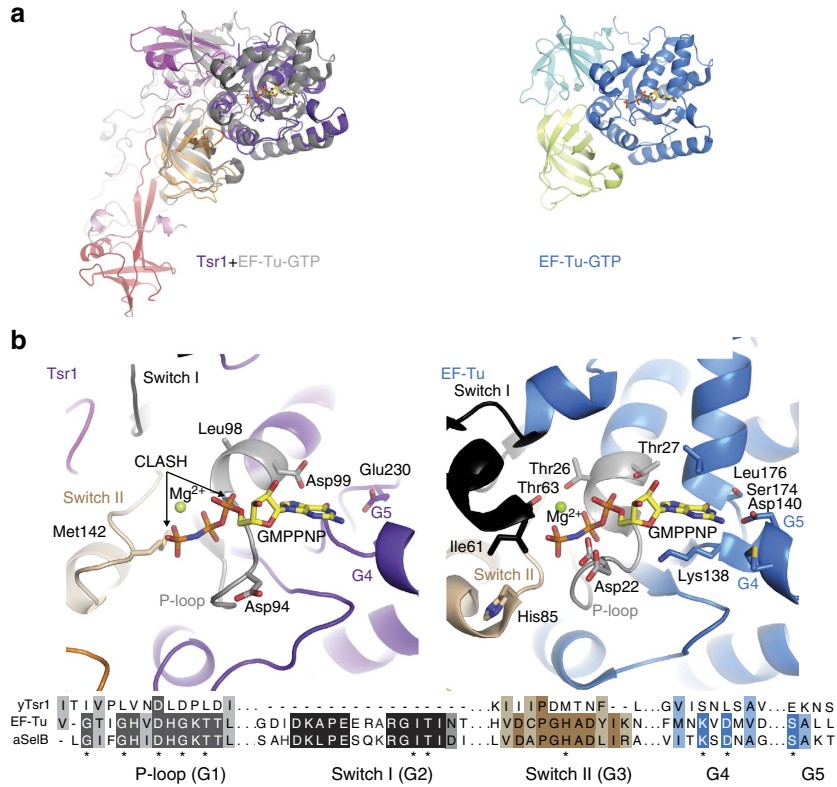

**Figure 2 | Tsr1 lacks a functional GTPase active site.** (**a**) Comparison of Tsr1 with the crystal structure of EF-Tu in the GTP-bound state (PDBid 1exm). A cartoon representation of Tsr1 is shown superposed with EF-Tu in grey on the left, while a cartoon structure of EF-Tu is shown in the same orientation on the right. EF-Tu domains I–III are coloured blue, teal and green, respectively. The GTP analogue GMPPNP is shown as sticks. (**b**) A close-up view of the active site of EF-Tu (right) and the equivalent site of Tsr1 with GMPPNP from EF-Tu superposed. Clashes between GMPPNP and Tsr1 are indicated with arrows (left). Sequences for characteristic motifs of GTPases are shown underneath, derived from a structure-based sequence alignment. In both sequence and structures, switch I is black, switch II is wheat and the P-loop is grey.

N-terminal region, which crosslink to many lysine residues on the 'back' face of the protein (Supplementary Fig. 3b). Common to both samples are a series of crosslinks between the poorly defined loop region between residues 312 and 335 in domain II and a loop region centred around residue 130 in domain I (Supplementary Fig. 3b). Crosslinks that are specific to Tsr1$_{\Delta N \Delta loop}$ are primarily associated with the amino terminus of the protein (Supplementary Fig. 3c).

**Tsr1 is a structural mimic of translational GTPases.** On the basis of three-dimensional structural searches with DALI, domains I–III together show structural similarity to translational GTPases of the EF-Tu family (Fig. 2a, Z-score is 15.7, 18% identity)[16]. This similarity is not easily identifiable at the sequence level because of the large insertion segment in domain II. Furthermore, structural comparison allows the identification of equivalent sequences and secondary structure elements in the G-domain, which are difficult to align correctly using sequence data alone. Small GTPases are characterized by five sequence elements that are critical either for binding guanine nucleotides or for GTP hydrolysis[17]. Comparison of Tsr1 with EF-Tu shows that Tsr1 has a domain arrangement similar to EF-Tu in its GTP-bound, active form (Fig. 2a)[18]. However, all five of the characteristic G-protein sequence signatures are disrupted in Tsr1 (Fig. 2b). We conclude that Tsr1 is neither a catalytically active GTPase nor a guanine nucleotide-binding protein, consistent with previous functional studies[12,14].

The 'chalice'-like structure displayed by Tsr1 has previously been described in other translational GTPases. The closest structural relative to full-length Tsr1 is archaeal SelB (aSelB, Dali

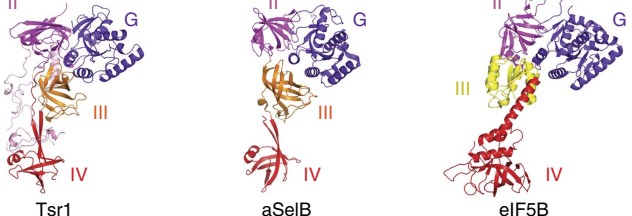

**Figure 3 | Tsr1 is related to translational GTPases.** Tsr1, aSelB (PDBid 9ac4) and eIF5B from budding yeast (PDBid 3wbk) are shown in a similar orientation after structural alignment. Equivalent domains are coloured according to the schema in Fig. 1. Domain III of eIF5B (yellow) has a different topology to domain III in Tsr1 and aSelB.

Z-score is 15.0, 17% identity), which is required for selenocysteine incorporation (Fig. 3)[19]. SelB-related proteins are also found in bacteria and eukaryotes, but bacterial SelB proteins differ in their domain architecture at the C terminus[20]. SelB proteins deliver selenocysteinyl-tRNAs to the ribosome in response to *cis*-acting sequences within an mRNA, allowing selenocysteine incorporation at specific stop codons[21]. Comparison of Tsr1 and aSelB from *Methanoccus maripaludis* shows that the overall architecture and domain topology is similar between these two proteins and that the primary difference is the long insertion in domain II of Tsr1 (Fig. 3)[19]. While individual domains of aSelB and Tsr1 have similar folds (r.m.s.d. values of 2.7, 1.8, 2.3 and 3.0 Å for domains I–IV, respectively), domain II is rotated relative to domains I and III, such that superposing all domains gives an r.m.s.d. value of 4.2 Å over 471 Cα atoms.

The overall shape of Tsr1 also resembles translation initiation factor eIF5B (Fun12p in yeast), which mediates pre-40S:60S interactions during a late, quality control step in 40S maturation (Fig. 3)[12,13]. eIF5B, however, differs in its domain architecture: domain III is a mixed α–β domain that is followed by a long α-helix. The C-terminal β-barrel domain of eIF5B has a similar topology to domain IV of Tsr1 and aSelB but is oriented differently (Fig. 3)[22,23]. The similarity in the shape of these proteins, despite their topological differences, suggests that placement of the C-terminal domain distant from the core of the protein is an important functional feature. Indeed, in structures of eukaryotic ribosomal initiation complexes, the C-terminal domain of eIF5B lies close to the tRNA-binding sites on the 40S subunit, while the GTPase domain interacts with the 60S subunit[24,25].

**Tsr1 and eIF5B binding sites overlap in pre-40S particles.** To understand how Tsr1 interacts with pre-40S particles, we reasoned that ribosome-interacting surfaces on Tsr1 are likely to be both conserved and positively charged to interact with the RNA core of the particle. Surface analysis of electrostatic potential of Tsr1 reveals clear differences between the 'front' and 'back' surfaces of the protein (Figs 1c and 4a)[26]. The front surface is rich in glutamate and aspartate residues creating an extensive, negatively charged surface. In contrast, the back surface of Tsr1 has several clusters of lysine and arginine residues, and is mainly positively charged (Fig. 4a).

Surface conservation of Tsr1 was calculated using the CONSURF server, utilizing aligned sequences from fungi and metazoans[27]. Highest levels of surface conservation were found in domain IV and on the back surface of Tsr1, broadly corresponding to the positively charged surface (Fig. 4b). Interestingly, the front face of the GTPase-like domain is the least conserved part of the protein, consistent with the degenerated nature of the active site.

We modelled the binding site of Tsr1 using cryo-EM maps of late-pre-40S particles from wild-type and Tsr1-depleted yeast[15]. Using volume-fitting operations in Chimera[28], we fitted Tsr1 into a volume corresponding to the largest calculated difference between the two maps, which was previously assigned as Tsr1 density (Fig. 5a and Supplementary Fig. 4a)[15]. The best fit was obtained with the conserved, positively charged surface of Tsr1 towards the pre-40S particle and with the degenerated active site of Tsr1 facing outwards. This orientation placed the conserved domain IV towards the mRNA-binding channel of the pre-40S particles (Fig. 5a). This orientation also places the negatively charged surface of Tsr1 on the outside of the particle, on the surface that would interact with the 60S subunit in mature

ribosomes (Fig. 5a). This placement of the acidic face suggests that yeast Tsr1 may prevent association of mature 60S subunits by electrostatic repulsion of the negatively charged RNA backbone. Overall, the placement of Tsr1 supports the model that late pre-40S ribosome biogenesis factors act to block to premature association with mature 60S particles[15,29].

The model of Tsr1 bound to the pre-40S particle was superposed with eIF5B-bound 40S subunits from a mammalian pre-initiation complex[25]. Comparison of the Tsr1 and eIF5B positions in these superposed structures reveals substantial overlap between the binding sites of Tsr1 and eIF5B (Fig. 5b and Supplementary Fig. 4b). eIF5B occupies a position on 40S subunits that is used by several GTPases during translation initiation, elongation and termination. The orientations of both Tsr1 and eIF5B result in their G-domains projecting out from the particle. In the case of eIF5B, the G-domain makes critical interactions with the 60S subunit while the C-terminal domain is placed close to the mRNA-binding channel. The orientation of Tsr1 binding is distinct from eIF5B, in that it appears to interact intimately with parts of pre-40S particle known as the head and beak. This placement of Tsr1 occludes part of the eIF5B-binding site (Fig. 5b and Supplementary Fig. 4b). The positioning of Tsr1 and eIF5B suggests binding of these two proteins is mutually exclusive during pre-40S maturation.

**The Tsr1 binding site overlaps Rio1 crosslinking sites.** Previous crosslinking and cDNA analysis (CRAC) of Rio1 binding on 18S rRNA indicate that Rio1 associates with bases around nucleotides 553–583, 1,094–1,106 and 1,631–1,643 (ref. 10). These sites are located in the mRNA-binding channel of mature 40S subunits. We modelled a yeast 40S subunit to the pre-40S particle map[24]. In this superposition, the Rio1 crosslinked bases appear to be occluded by the presence of Tsr1 (Fig. 5c). Furthermore, there is an overlap between Rio1 CRAC sites and Tsr1 CRAC sites[29] (Fig. 5d). Several Tsr1 CRAC sites mapped to bases 1,061–1,134, indicating an overlap with Rio1 binding at 1,094–1,106. Further Tsr1 sites are located at 1,143–1,146 and 1,767–1,769. In mature 40S subunits, bases 1,145–1,146 pair with 1,633 and 1,635, respectively, while base 1,767 pairs with 1,636 indicating that these two sites may also be in proximity in the pre-40S particles (Fig. 5d). The overlap between the Tsr1- and Rio1-binding sites and the apparent occlusion of the Rio1 site by Tsr1 suggest that loss of Tsr1 is also a pre-requisite for Rio1 association with pre-40S particles. This is consistent with analysis of pre-40S particles purified using Rio1 tandem affinity purification that showed very low levels of Tsr1 (as well as other 'early' factors such as Enp1, Ltv1, Rio2 and Dim1)[10].

## Discussion
Tsr1 is a key nucleo-cytoplasmic pre-40S ribosome biogenesis factor that is essential for the final maturation of 20S rRNA to 18S rRNA[9,10,14]. Our structural studies reveal that Tsr1 is structurally related to translational GTPases such as EF-Tu and aSelB, and adopts a conformation similar to that of the activated, GTP-bound state of EF-Tu. However, Tsr1 lacks all the conserved motifs required for guanine nucleotide binding and GTP hydrolysis, and is therefore a structural mimic of translational GTPases rather than a functional enzyme.

Modelling of Tsr1 into cryo-EM maps of pre-40S particles places it in a position that occludes the mRNA-binding channel around the A-site and overlaps with the binding site of eIF5B, a translational GTPase that is required during translation initiation to mediate 60S joining. This orientation places the conserved C-terminal domain of Tsr1 in close proximity to the head and beak regions of the pre-40S particle. Tsr1 does not bind

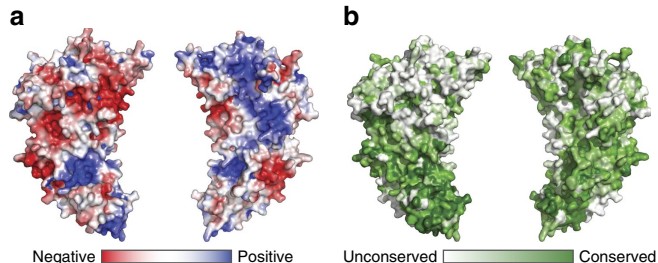

**a** **b**

Negative ▬▬▬ Positive Unconserved ▬▬▬ Conserved

**Figure 4 | Surface properties of Tsr1. (a)** Electrostatic surface calculation for an all-side chain model of Tsr1, calculated using adaptive Poisson-Boltzman solver (APBS). Negatively charged surfaces are red ($-4.5\,kTe^{-1}$) and positively charged surfaces are blue ($+4.5\,kTe^{-1}$). **(b)** Conservation of surface residues as calculated by CONSURF, showing non-conserved surfaces in white and a gradient of conservation in green.

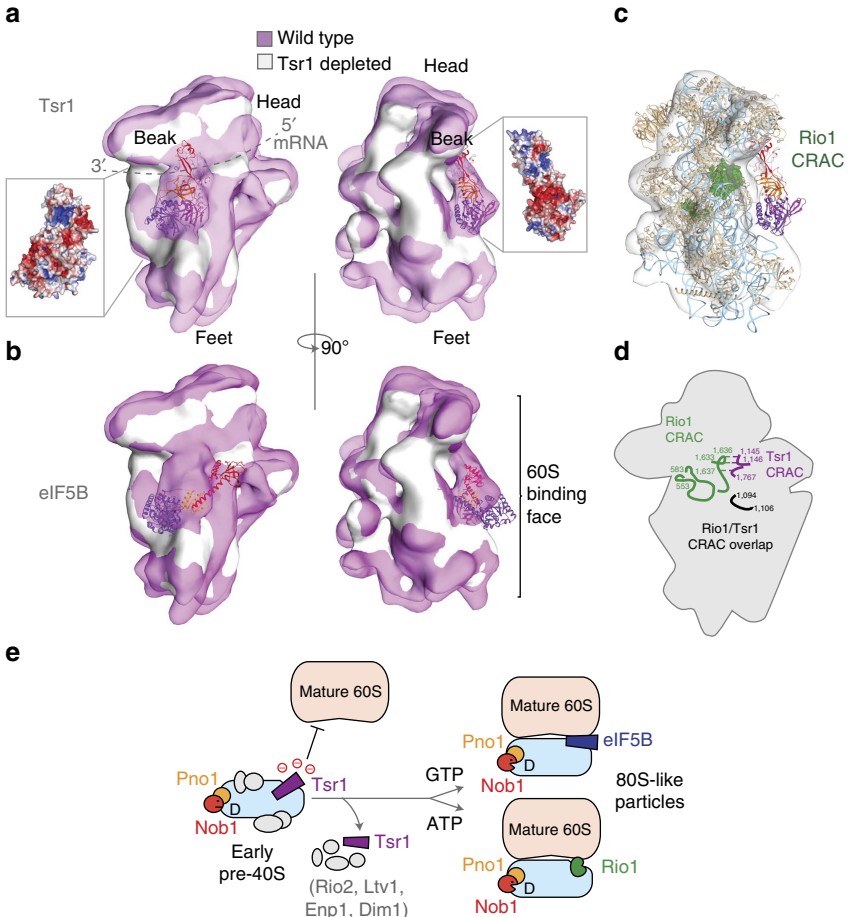

**Figure 5 | Tsr1 excludes binding of eIF5B and Rio1.** (**a**) The Tsr1 structure was fitted into aligned, previously published cryo-EM maps of pre-40S particles derived from wild-type (transparent magenta, EMD1926) and Tsr1-depleted (opaque white, EMD1927) yeast cells. The particle is viewed from the mRNA-binding surface (left) and a 90° rotation (right) to show the positioning of Tsr1 with respect to the 60S interaction surface. Tsr1 is shown with the front view rotated ~180° in the plane of the page with respect to Fig. 1c. Inset figures indicate the electrostatic surface of Tsr1 when fitted into the particle. (**b**) A 40S structure from a mammalian pre-initiation complex, including eIF5B (PDBID 4UJD), was fitted to EMD1926. Proteins, rRNA and tRNA from the pre-initiation complex are hidden, with the exception of eIF5B, for the sake of clarity. The orientations are as in **a**. The face of the 40S that interacts with the 60S is indicated with a bracket. (**c**) Side view of Tsr1 fitted to cryo-EM particles with 40S subunit from yeast fitted into the map (PDBID 4V8Y). Bases previously identified as Rio1 crosslinking sites by CRAC are coloured green and shown as space-fill. (**d**) Overview of the arrangement of Rio1 (green), Tsr1 (purple) and common (black) CRAC sites. (**e**) Tsr1 binding is mutually exclusive with binding of eIF5b and Rio1, and with 60S subunit joining. We speculate that negative charges on the surface of Tsr1 prevent premature 60S joining. Displacement of Tsr1, along with other early and intermediate pre-40S assembly factors (Rio2, Ltv1, Enp1 and Dim1), is required for late pre-40S maturation. Following removal of Tsr1, eIF5B-binding promotes GTP-dependent 60S subunit association, while Rio1 is predicted to promote ATP-dependent association. In both cases, formation of these 80S-like particles may constitute a quality control step, verifying that the pre-40S particle is close to the mature structure, before final 3′ processing of the 18S rRNA by the endonuclease Nob1.

precisely at the translational GTPase site, perhaps reflecting its evolution from a transiently interacting enzyme to a stably bound biogenesis factor.

eIF5B has also been shown to mediate 60S joining to pre-40S particles during a 'translation-like' cycle that activates the final cleavage of 20S rRNA at site D, to produce mature 18S rRNA[12,13]. Late cytoplasmic maturation of pre-40S particles is driven by both GTPase (associated with eIF5B) and ATPase (associated with Rio1) activities that do not appear to be cooperative[10,12]. Our observation that Tsr1 is also likely to occlude the Rio1-binding site suggests that loss of Tsr1 from pre-40S particles is a pre-requisite for late maturation steps on both pathways (Fig. 5e).

We propose that Tsr1 association with pre-40S particles performs multiple functions. It blocks the mRNA-binding channel, prevents 60S association through steric and electrostatic mechanisms, and blocks the binding sites for eIF5B and Rio1.

Since Tsr1 is not catalytically active these data suggest that, rather than acting as a molecular switch, Tsr1 is a molecular gatekeeper that must be removed before key sites can be accessed to catalyse the final stages of pre-40S maturation. The proximity of the eIF5B- and Rio1-binding sites further suggests that the two routes to final 40S maturation catalysed by these proteins may be mutually exclusive (Fig. 5e)[10,24]. Future functional studies will reveal the events required to release pre-40S particles from the molecular grip of Tsr1.

## Methods

**Protein purification and crystallization.** The coding sequence for *S. cerevisiae* Tsr1 was cloned and expressed in *E. coli* using a pET-based vector with an N-terminal hexahistidine tag and 3C protease cleavage site. Cells were expressed at 37 °C and subsequently cooled to 18 °C when cultures reached an OD$_{600}$ of 0.6–1.0. Protein expression was induced with 0.2 mM isopropyl-β-D-thiogalactoside and carried out overnight at 18 °C. In-frame deletions to the Tsr1 sequence to create the

Tsr1 ΔN46Δ410–476 construct were carried out by inverse PCR mutagenesis to generate the internal loop deletion. Priming downstream of the start codon generated the N-terminal deletion during PCR amplification of the cDNA sequence followed by ligation-independent cloning into a pET-based vector using the T4 polymerase system. Tsr1 was purified using batch binding of clarified cell lysate to Ni$^{2+}$-NTA in 20 mM Tris-HCl (pH 8.0), 500 mM NaCl, 10 mM imidazole (pH 8.0) and 1 mM β-mercaptoethanol, and elution in the same buffer supplemented with 500 mM imidazole. Eluted protein was dialysed overnight at 4 °C into 20 mM Tris-HCl (pH 8.0), 600 mM NaCl, 5% glycerol and 1 mM dithiothreitol in the presence of glutathione S-transferase-tagged rhinovirus 3C protease to remove the hexahistidine tag. The sample was subsequently concentrated and separated on a superdex S200 HiLoad (16/60) size-exclusion column (GE Healthcare) in 20 mM Tris-HCl (pH 8.0), 800 mM NaCl, 5% glycerol and 1 mM dithiothreitol, and concentrated to 8 mg ml$^{-1}$ for crystallization. Optimal conditions for crystal growth were 8–14% w/v polyethylene glycol 3350 and 0.18–0.30 M sodium malonate (pH 6.0) at 18 °C using 24-well sitting drop plates.

**Crystal soaking and data collection.** To make TaBr cluster-derivatized crystals, native crystals were transferred to a stabilization buffer containing 1–2 mM TaBr cluster (Jena Biosciences) and 14% polyethylene glycol 3350 and 0.3 M sodium malonate (pH 6.0) for 1 h. Crystals were then cryoprotected in stabilization buffer supplemented with 30% glycerol and flash cooled in liquid nitrogen. Data were collected at three wavelengths (1.2548 (9880.9), 1.2552 (9877.4) and 1.2536 Å (9890.0 eV) for peak, inflection and high-energy remote, respectively) at beamline I02 (Diamond Light Source, Didcot, UK). A higher-resolution native data set was collected at I03 ($\lambda = 0.97$ Å), and a series of native data sets at $\lambda = 1.77$ Å were collected at I04 and I02 for sulfur-SAD (S-SAD) phasing. Where possible, multiple data sets for S-SAD phasing were collected from the same crystal after translating its position and using a kappa offset of 10°. For each data set, 360° of images were collected using an oscillation angle of 0.15° per image and beam transmission levels between 10 and 20%. Data sets were indexed and integrated using XDS[30] and scaled using SCALA[31]. Native data sets for S-SAD phasing were analysed using BLEND[32], and data sets were selected for scaling based on their degree of isomorphism (Linear Cell Variation of <1%).

**Model building and refinement.** Multiple wavelength anomalous dispersion data from the TaBr-soaked crystals were used to calculate the heavy metal substructure to 7 Å using the SHELX package[33] and HKL2MAP (ref. 34). Phases were refined using SHARP and extended to 4.5 Å using automated solvent-flattening routines[35]. Phase information from the TaBr data set was then used to calculate the positions of sulfur atoms from the native S-SAD data set and these peak positions were used in PHENIX AutoBuild to calculate non-crystallographic symmetry (NCS) operators for the four molecules in the asymmetric unit[36,37]. Solvent flattening and NCS averaging in PHENIX allowed the calculation of an interpretable electron density map at 4.2 Å, and a S-SAD anomalous difference map was used to aid sequence interpretation. Model building was carried out in COOT[38] and O (ref. 39). Refinement was carried out in BUSTER[40]. After initial rounds of model building and refinement using the experimentally phased TaBr map, molecular replacement, using PHASER[41], was used to locate the model in the 3.6-Å native data set. The final model was validated using Molprobity[42]. In the structure, 92.1% of the residues had Ramachandran angles in favoured regions, with 7.1% in allowed regions and 0.8% in disallowed regions.

Fitting the Tsr1 model in to cryo-EM maps was carried out using volume operations in CHIMERA[28]. Maps from wild-type (EMD1927) and Tsr1-depleted (EMD1926) pre-40S particles were aligned manually and fitted to give optimal overlap[15]. The depletion mutant map was resampled on the wild-type map and a volume difference was calculated. Tsr1 coordinates were manually aligned to a large difference peak on the 60S-interacting face of the pre-40S particles and the best fit calculated in two different orientations. The orientation with the fewest atoms outside of the density was retained for comparison with eIF5B. Coordinates for the 40S particle and eIF5B from 80S pre-initiation complexes (PDBIDs 4UJD, 4V8Y) were extracted and fitted to the map corresponding to the wild-type pre-40S particles (EMD1927) using manual alignment followed by optimized fitting[24,25]. The position of eIF5B relative to the position where Tsr1 was fitted to difference density was compared.

**Yeast complementation assay.** The chromosomal *TSR1* gene was fused to an N-terminal triple hemagglutinin (HA) and placed under the control of the $P_{GAL1}$ promoter in a BY4741 background using pYM-24 plasmid as previously described[43]. After selection on media containing nourseothricin and galactose, correct integration was verified by colony PCR. The $P_{GAL}::HA-TSR1$ cells, together with the corresponding wild-type control, were transformed with either pRS313 or pRS313-TSR1$_{ΔNΔloop}$ vectors. Cells were grown to logarithmic phase in synthetic galactose medium without histidine. Tenfold serial dilutions were spotted on synthetic media without histidine and containing either 2% glucose or 2% galactose. Images were taken on the second day of incubation at 30 °C.

Primers used for pYM-24 amplification were TSR1_S1, 5′-tatttgttagttgaagagcggt agtttacgcaggcatcagaatgcgtacgctgcaggtcgac-3′; and TSR1_S4, 5′-agatttgtgtccgttttta atgatgacctgtgtgaatgacctgccatcgatgaattctctgtcg-3′. Primers used for verification were

TSR1_chk1, 5′-gttctgctcatctcatcaccag-3′; TSR1_chk2, 5′-gcacctttagaagcatgtttg-3′; pYM_chk1_2, 5′-gtcgacctgcagcgtacg-3′; and pYM_chk2_2, 5′-cgacagagaattcatcg atg-3′.

**Data Availability.** Structural data have been deposited in the PDB with the following accession code: 5IW7. Cross-linking mass spectrometry data have been deposited in PRIDE with the following accession code: PXD004074.

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

## Acknowledgements

We thank the Xtalpod facility for crystallization screening and beam-line scientists at Diamond Light Source for support during data collection. We thank the Edinburgh Protein Production Facility for support with protein purification and biophysical characterization. We thank A.A. Jeyaprakash and Fulvia Bono for critical reading of the manuscript. A.G.C. and U.J. were supported by the Medical Research Council (G1000520/1); D.T., J.R. and U.M.M. were supported by the Wellcome Trust (077248, 103139 and 093851); V.S. was supported by the SNSF (P2EZP3_159110). The Wellcome Trust Centre for Cell Biology is supported by core funding and instrument grants (092076, 095822 and 108504). Crystallisation facilities were supported by a Wellcome Trust/University of Edinburgh Institutional Strategic Support Fund.

## Author contributions

U.M.M. and U.J. purified and crystallized the protein; U.M.M. and A.G.C. solved the crystal structure; V.S. carried out the yeast complementation assay; Z.A.C. and J.R. provided crosslinking/MS data and analysis; A.G.C. and D.T. wrote the manuscript.

## Additional information

**Competing financial interests:** The authors declare no competing financial interests.

