## [Peer Review File · Nature Communications]

Reviewer #1 (Remarks to the Author):

McCaughan et al. have determined the crystal structure of yeast Tsr1, a factor involved in ribosome biogenesis, and made the interesting observation that it is a catalytically inactive structural mimic of translational GTPases, being most similar to archaeal SelB, but also resembling eIF5B, which mediates subunit joining during translation initiation and functions during a late quality control step in ribosomal 40S subunit maturation. Docking the Tsr1 structure onto the cryoEM structure of pre-40S subunits suggested that Tsr1 occludes the eIF5B binding site; correlating the binding site with prior cross-linking analyses of sites on the pre-40S subunit bound by the kinase Rio1 suggested that Tsr1 must dissociate for Rio1 to be able to bind. Taken together, these observations suggest that Tsr1 functions as a gatekeeper that regulates access to the pre-40S subunit of factors involved in late stages in its maturation. These observations are novel, interesting and clearly presented, and the manuscript merits publication.

MINOR COMMENTS.

1. p6, line 9-12. The authors compared binding of Tsr1 to pre-40S subunits to binding of eIF5B to 40S subunits (p6, last 2 lines) using a structure (Ref. 23) in which domain IV of eIF5B was incorrectly fitted (see PMID 25350701 and 25064512 for an explanation of the error). It would be more appropriate to use the structure determined by Yamamoto et al (PMID 25064512).

Reviewer #2 (Remarks to the Author):

Here McCaughan et al. describe the crystal structure of Tsr1, a eukaryotic ribosome assembly factor. The atomic structure of this protein, which is structurally related to translational GTPases but catalytically inactive, is later docked into cryo-electron density maps obtained by Strunk et al. (ref. 15). The authors hypothesize that the presence of Tsr1 acts as quality control by preventing the association of other ribosome assembly factors or the 60S ribosomal subunit to the pre-40S particle.

While the architecture of Tsr1 could be predicted using homology modeling servers (Phyre2 and HHpred, which both selected SelB as structural template), the crystal structure of Tsr1 will be very useful for the field. The position of Tsr1 within the pre-40S particle agrees well with data previously obtained by Strunk et al. (ref. 15). The notion that ribosome assembly factors act by steric hindrance has previously been hypothesized but is up to this point hard to verify biochemically.

Regarding the methodology, McCaughan et al. have used an elegant combination of heavy atom cluster and native sulfur-SAD phasing to obtain the atomic structure of Tsr1.

There are only two parts where additional information from the authors would be useful for a revision of this manuscript.

1. Given the introduction of CC1/2 in crystallography (Diederichs & Karplus, 2013), I was surprised to see that the native dataset was still cut off at 3.6 Angstroms with an I/signal of 2.9 and a CC1/2 of 89.5% in the highest resolution shell. To understand the authors' reasoning, it would be very helpful to see the output after scaling, which includes more high resolution data together with comparative maps at the two resolutions.

1. The pdb reports that I received as part of this manuscript seem to list a completely different crystal form and protein structure. The updated pdb report listing the reported data would be very helpful for a revision of the manuscript.

References

Diederichs, K., & Karplus, P. A. (2013). Better models by discarding data? *Acta Crystallographica. Section D, Biological Crystallography*, 69(Pt 7), 1215-1222.
<http://doi.org/10.1107/S0907444913001121>

Reviewer #3 (Remarks to the Author):

The manuscript by McCaughan et al. reports the crystal structure of the ribosome biogenesis factor Tsr1 at 3.6Å resolution. Tsr1 shares similarities with GTPases such as Ef-Tu, but it is not enzymatically active. A poorly conserved region, which was removed to obtain the crystal structure, is shown to be not essential for cell viability. Furthermore docking into previously published cryo-EM maps is performed and the authors suggest that Tsr1 blocks binding of eIF5B and Rio1 as well as premature binding of 60S subunit. Overall, the manuscript is a useful contribution to the current understanding

of ribosome biogenesis. It is well written and clear, however a number of issues need to be resolved or clarified.

Page 3

...and to discover whether this protein is a cryptic GTPase, we determined the crystal structure of yeast Tsr1.

The term cryptic GTPase is ambiguous here. The authors themselves note that neither GTP binding nor hydrolysis has been observed before. Please reword or clarify.

Page 5

The closest structural relative to full-length Tsr1 is archaeal SelB (aSelB)

If this structural homology was determined with DALI then the authors should describe it in the material and methods section and cite the relevant publication [Holm L, Rosenström P (2010) Dali server: conservation mapping in 3D. Nucl. Acids Res. 38, W545-549 / PMID 20457744]. In addition it would be interesting to the reader if the authors would report the sequence identity/similarity to those structural relatives. This might further help demonstrating that there is high degree of structural similarity despite low sequence similarity.

Page 7

...and GTP hydrolysis and is therefore a dead enzyme.

This should be reworded as it implies that some catalytic activity had been present during evolution, which might be true or not. There are "NTPases" that truly contain all residues required for binding but lack the critical residues for hydrolysis 'by nature' and can be considered 'dead enzymes'. Due to lack of GTP binding, Tsr1 is at best described as a structural mimic/placeholder (in terms of shape/topology) of translational GTPases, but similar to other proteins it has no enzymatic activity.

Page 8 and elsewhere

Tris.HCl pH 8.0

Should be corrected to Tris-HCl or Tris/HCl

Page 4

S. cerevisiae Tsr1 was expressed as an N-terminal GST fusion protein in *E. coli*.

Page 8

Protein purification and crystallization

At the beginning of the results section it is mentioned that the protein was produced as a GST fusion protein. The material and methods section contains additional information about presence of a 3C protease cleavage site. However the purification strategy described lacks the step where the fusion protein was cleaved with 3C protease. Please clarify.

Page 8

Protein purification and crystallization

The authors should report the temperature and nature (sitting/hanging drop) of the crystallization trials.

Page 8

Induction with 0.2 mM IPTG was carried out overnight at 18{degree sign}C.

This should be reworded, since protein expression was carried out overnight at 18{degree sign}C and not induction.

Page 9

Data were collected at three wavelengths (1.254 Å, 1.255 Å, 1.254 Å for peak, inflection and high energy remote, respectively) at beamline I02 (Diamond Light Source, Didcot, UK).

The wavelengths reported seem to have been mixed up; high energy remote and peak are identical, please correct.

Page 9

Reference 32 is misplaced.

Page 11

Table1

The wavelengths for the datasets should be reported in Table 1. CC1/2 and I/σ(I) values indicate that the data sets may contain useful data beyond the currently chosen resolution cutoffs. Have the

authors compared the current electron density map against one calculated from a more 'aggressive' cutoff for the native data set?

Figure 2b

It might be easier to appreciate the incompatibility of GTP-binding if the GTP/GMPPNP from the EF-Tu comparison would be shown here (transparent/sticks), optionally with clashes that might occur.

Figure 4a

Electrostatic surface potential should be calculated with APBS or Delphi. The range should be indicated either in the figure or in the figure legend.

One of the main points of the paper is the presence of a highly acidic surface on Tsr1, which based on the docking within cryo-EM maps, faces the 60S subunit and thereby prevents subunit joining. The authors should visualize this, for example by showing charge distribution in the cryo-EM docking and indicating the direction of 60S subunit binding. Alternatively they may integrate figure 4a into figure 5 and chose similar or same orientation to ease the identification of Tsr1 and its charge within the cryo-EM maps. This (charge repulsion) may also be added to the model in figure 5e.

Supplement methods

Page 4

Figure S2b

The global superposition of all four chains should use four slightly different colors.

We thank the reviewers for their constructive comments. Please find below our point-by-point response to the issues that they raised.

Reviewer 1

p6, line 9-12. The authors compared binding of Tsr1 to pre-40S subunits to binding of eIF5B to 40S subunits (p6, last 2 lines) using a structure (Ref. 23) in which domain IV of eIF5B was incorrectly fitted (see PMID 25350701 and 25064512 for an explanation of the error). It would be more appropriate to use the structure determined by Yamamoto et al (PMID 25064512).

We have remade Fig 5b using the 40S subunit and eIF5B structure from Yamamoto *et al.* The supplemental figure, Fig. S4b, which accompanies Fig. 5b, has also been remade with this model. We have retained our original fit to the yeast 40S subunit from Fernandez *et al*/ in Fig. 5c as this part of the structure does not contain the mis-modeled region, yet it relies on a comparison of yeast CRAC data with yeast pre-40S cryoEM maps.

Reviewer 2

While the architecture of Tsr1 could be predicted using homology modeling servers (Phyre2 and HHpred, which both selected SelB as structural template), the crystal structure of Tsr1 will be very useful for the field. The position of Tsr1 within the pre-40S particle agrees well with data previously obtained by Strunk et al. (ref. 15). The notion that ribosome assembly factors act by steric hindrance has previously been hypothesized but is up to this point hard to verify biochemically.

1. Given the introduction of CC1/2 in crystallography (Diederichs & Karplus, 2013), I was surprised to see that the native dataset was still cut off at 3.6 Angstroms with an $I/\sigma I$ of 2.9 and a CC1/2 of 89.5% in the highest resolution shell. To understand the authors' reasoning, it would be very helpful to see the output after scaling, which includes more high resolution data together with comparative maps at the two resolutions.

We set our resolution limit conservatively at 3.6 Å as the merging statistics for our native dataset are considerably worse beyond this limit. However, as reviewers #2 and #3 note, there may still be useful data beyond this limit as indicated by the $I/\sigma I$ and the CC(1/2) (Reviewer Fig.1).

SUBSET OF INTENSITY DATA WITH SIGNAL/NOISE ≥ -3.0 AS FUNCTION OF RESOLUTION													
RESOLUTION LIMIT	NUMBER OF REFLECTIONS			COMPLETENESS OF DATA	R-FACTOR observed	R-FACTOR COMPARED expected	I/SIGMA	R-meas	CC(1/2)	Anomal Corr	SigAno	Nano	
	OBSERVED	UNIQUE	POSSIBLE										
10.00	19548	2230	2261	98.6%	3.5%	3.6%	19542	51.52	3.7%	99.9*	-8	0.779	1586
6.00	74398	7545	7545	100.0%	6.1%	6.0%	74394	29.88	6.4%	99.9*	-3	0.794	6330
5.00	68932	6907	6907	100.0%	11.9%	11.9%	68931	17.64	12.5%	99.6*	-2	0.797	6050
4.00	156179	15437	15439	100.0%	20.5%	20.4%	156178	11.04	21.6%	99.2*	-2	0.794	13866
3.80	53054	5196	5196	100.0%	54.6%	54.8%	53054	4.70	57.4%	95.0*	-3	0.746	4720
3.60	65789	6456	6457	100.0%	88.7%	89.0%	65789	2.98	93.3%	90.1*	-2	0.723	5896
3.50	37935	3768	3770	99.9%	137.6%	138.3%	37930	1.94	144.9%	78.8*	0	0.705	3445
3.40	581	251	4242	5.9%	263.6%	256.1%	493	0.32	321.5%	15.5	-18	0.576	31
total	476416	47790	51817	92.2%	12.9%	12.9%	476311	14.31	13.6%	99.9*	-2	0.771	41924

Reviewer Fig. 1 Scaling statistics of native data from XSCALE.

We recalculated the maps after re-scaling the data to include reflections up to 3.5 Å. Reviewer Fig.2 shows part of the density taken from a similar region as

shown in Fig S2a, but with a slightly larger region of the structure shown and a small rotation around the vertical axis to better show the alpha helix. Overall, including more data has not increased the detail in the map. However, in solvent channels there are additional noise peaks in the 3.5 Å map. As including higher resolution data does not significantly improve the maps and introduces some noise, we think that the resolution cutoff of 3.6 Å reflects the level of detail that we see in our maps.

Reviewer Fig. 2. Comparison of 2Fo-Fc maps, contoured at 1σ , calculated from data to 3.6 Å (pink) compared with a similar map calculated from data to 3.5 Å (blue).

2. The pdb reports that I received as part of this manuscript seem to list a completely different crystal form and protein structure. The updated pdb report listing the reported data would be very helpful for a revision of the manuscript.

We have uploaded an updated report.

Reviewer 3

1. Page 3
...and to discover whether this protein is a cryptic GTPase, we determined the crystal structure of yeast Tsr1.
The term cryptic GTPase is ambiguous here. The authors themselves note that neither GTP binding nor hydrolysis has been observed before. Please reword or clarify.

We have reworded “ ... and to discover whether this protein is a functional GTPase, ...” to remove the ambiguity.

2. Page 5

The closest structural relative to full-length Tsr1 is archaeal SelB (aSelB). If this structural homology was determined with DALI then the authors should describe it in the material and methods section and cite the relevant publication [Holm L, Rosenström P (2010) Dali server: conservation mapping in 3D. Nucl. Acids Res. 38, W545-549 / PMID 20457744]. In addition it would be interesting to the reader if the authors would report the sequence identity/similarity to those structural relatives. This might further help demonstrating that there is high degree of structural similarity despite low sequence similarity.

We originally identified the structural similarity between SelB and Tsr1 prior to solving the crystal structure by using a threading approach with the Phyre server. As noted by Reviewer #2, this structural similarity could also be detected with HHPred. For completeness we have conducted a Dali search and found that EF-Tu and SelB are indeed the highest hits on the list with several other translational GTPases identified as well. We have included the reference to Holm and Rosenström and indicated the Dali Z-scores and percentage sequence identity in the text.

3. Page 7

...and GTP hydrolysis and is therefore a dead enzyme.

This should be reworded as it implies that some catalytic activity had been present during evolution, which might be true or not. There are "NTPases" that truly contain all residues required for binding but lack the critical residues for hydrolysis 'by nature' and can be considered 'dead enzymes'. Due to lack of GTP binding, Tsr1 is at best described as a structural mimic/placeholder (in terms of shape/topology) of translational GTPases, but similar to other proteins it has no enzymatic activity.

We have reworded as follows:

"...Tsr1 lacks all the conserved motifs required for guanine nucleotide binding and GTP hydrolysis and is therefore a structural mimic of translational GTPases rather than a functional enzyme. "

4. Page 8 and elsewhere

Tris.HCl pH 8.0

Should be corrected to Tris-HCl or Tris/HCl

Corrected

5. Page 4

S. cerevisiae Tsr1 was expressed as an N-terminal GST fusion protein in *E. coli*.

Page 8

Protein purification and crystallization

At the beginning of the results section it is mentioned that the protein was produced as a GST fusion protein. The material and methods section contains

additional information about presence of a 3C protease cleavage site. However the purification strategy described lacks the step where the fusion protein was cleaved with 3C protease. Please clarify.

The differences in description of the affinity tag between the results and materials and methods section were an oversight that has been corrected. The protein was expressed as an N-terminal hexahistidine tag. Details of the 3C cleavage reaction, which was carried out after affinity purification, have been included in the materials and methods section.

6. Page 8

Protein purification and crystallization

The authors should report the temperature and nature (sitting/hanging drop) of the crystallization trials.

These details have been added in the text.

7. Page 8

Induction with 0.2 mM IPTG was carried out overnight at 18 {degree sign}C.

This should be reworded, since protein expression was carried out overnight at 18 {degree sign}C and not induction.

This has been changed to read "... Protein expression was induced with 0.2 mM IPTG and carried out overnight at 18°C".

8. Page 9

Data were collected at three wavelengths (1.254 Å, 1.255 Å, 1.254 Å for peak, inflection and high energy remote, respectively) at beamline I02 (Diamond Light Source, Didcot, UK).

The wavelengths reported seem to have been mixed up; high energy remote and peak are identical, please correct.

This has been corrected in the text and the corresponding energy at each wavelength has been added to further clarify.

9. Page 9

Reference 32 is misplaced.

This reference has been replaced with the more recent reference for AUTOSHARP (Vonrhein et al 2007).

10. Page 11

Table 1

The wavelengths for the datasets should be reported in Table 1.

Wavelengths for datasets have been reported in the materials and methods rather than in Table 1, in keeping with the requirements of the journal.

11. CC1/2 and I/σ(I) values indicate that the data sets may contain useful data beyond the currently chosen resolution cutoffs. Have the authors compared the

current electron density map against one calculated from a more 'aggressive' cutoff for the native data set?

This comment has been addressed above, see reviewer #2's comments.

12. Figure 2b

It might be easier to appreciate the incompatibility of GTP-binding if the GTP/GMPPNP from the EF-Tu comparison would be shown here (transparent/sticks), optionally with clashes that might occur.

Figure 2b has been altered to show the position of the superposed GMPPNP with clashes indicated. The legend has been modified accordingly.

13. Figure 4a

Electrostatic surface potential should be calculated with APBS or Delphi. The range should be indicated either in the figure or in the figure legend.

Electrostatic surface potential in Fig. 4a has been recalculated with APBS and the range is indicated in the figure legend. A citation for APBS has been added in the text.

14. One of the main points of the paper is the presence of a highly acidic surface on Tsr1, which based on the docking within cryo-EM maps, faces the 60S subunit and thereby prevents subunit joining. The authors should visualize this, for example by showing charge distribution in the cryo-EM docking and indicating the direction of 60S subunit binding. Alternatively they may integrate figure 4a into figure 5 and chose similar or same orientation to ease the identification of Tsr1 and its charge within the cryo-EM maps. This (charge repulsion) may also be added to the model in figure 5e.

We have included new inset figures in Fig. 5a to show the view of electrostatic surface of Tsr1 when oriented in the particle. The direction of 60S binding is indicated with a bracket in Fig. 5b. We have also included an indication of the role of electrostatic repulsion in the model in Fig. 5e and further clarified this point in the text.

15. Supplement methods

Page 4

Figure S2b

The global superposition of all four chains should use four slightly different colors.

This figure has been altered to show each chain in a different colour.